# Raman evidence for pressure-induced formation of diamondene

Luiz Gustavo Pimenta Martins[1,7], Matheus J.S. Matos [2], Alexandre R. Paschoal[3], Paulo T.C. Freire[3], Nadia F. Andrade[4], Acrísio L. Aguiar[5], Jing Kong[6], Bernardo R.A. Neves [1], Alan B. de Oliveira[2], Mário S.C. Mazzoni[1], Antonio G. Souza Filho[3] & Luiz Gustavo Cançado[1]

Despite the advanced stage of diamond thin-film technology, with applications ranging from superconductivity to biosensing, the realization of a stable and atomically thick two-dimensional diamond material, named here as diamondene, is still forthcoming. Adding to the outstanding properties of its bulk and thin-film counterparts, diamondene is predicted to be a ferromagnetic semiconductor with spin polarized bands. Here, we provide spectroscopic evidence for the formation of diamondene by performing Raman spectroscopy of double-layer graphene under high pressure. The results are explained in terms of a breakdown in the Kohn anomaly associated with the finite size of the remaining graphene sites surrounded by the diamondene matrix. Ab initio calculations and molecular dynamics simulations are employed to clarify the mechanism of diamondene formation, which requires two or more layers of graphene subjected to high pressures in the presence of specific chemical groups such as hydroxyl groups or hydrogens.

[1] Departamento de Física, Universidade Federal de Minas Gerais, Belo Horizonte, MG 30123-970, Brazil. [2] Departamento de Física, Universidade Federal de Ouro Preto, Ouro Preto, MG 35400-000, Brazil. [3] Departamento de Física, Universidade Federal do Ceará, Fortaleza, CE 60455-900, Brazil. [4] Instituto Federal de Educação, Ciência e Tecnologia do Ceará, Tianguá, CE 62320-000, Brazil. [5] Departamento de Física, Universidade Federal do Piauí, Teresina, PI 64049-550, Brazil. [6] Department of Electrical Engineering and Computer Science, Massachusetts Institute of Technology, Cambridge, MA 02139, USA. [7] Present address: Department of Physics, Massachusetts Institute of Technology, Cambridge, MA 02139, USA. Correspondence and requests for materials should be addressed to L.G.C. (email: cancado@fisica.ufmg.br)

**D**iamond is the hardest and least compressible material[1–3] as well as the best bulk heat conductor[4]. In addition, it is chemically inert[5], highly refractive at optical wavelengths, and transparent to ultraviolet[6]. Unlike graphite, another bulk carbon allotrope that can easily exfoliate due to its layered hexagonal structure[7, 8], diamond does not present a stable two-dimensional (2D) counterpart to date, and this is mostly due to its tetrahedral structure. Nevertheless, many of the outstanding physical properties of graphene (the 2D version of graphite) rely on its dimensionality[7, 9] the same as with other 2D materials[10], such as phosphorene[11], silicene[12, 13], 2D transition metal dichalcogenides[14], and 2D transition metal carbides or nitrides[15]. Given the technological advances in the diamond thin-film production and applications[5, 16–21] the systematic realization of an atomically thin 2D diamond structure is highly desirable.

A first step was recently given by Barboza et al.,[22] who proposed and provided experimental evidence for the existence of a 2D diamond crystal formed when two or more layers of graphene are subjected to high pressures in the presence of chemical groups. With the assumption that the chemical groups are hydroxyl radicals, the compound was named diamondol, and was characterized as a 2D ferromagnetic semiconductor with spin polarized bands[22]. These unique properties, which arise from the periodic array of dangling bonds at the bottom layer, make diamondol a promising candidate for spintronics. Thus far, the existence of this 2D rehybridized carbon material has been demonstrated by electric force microscopy experiments, which

have monitored the charge injection into mono- and bi-layer graphene with increasing tip-force interaction and different water contents on the graphene surface[22].

Here we provide spectroscopic evidence of the formation of such a 2D-diamond structure, which we shall denote as diamondene, by performing Raman spectroscopy of double-layer graphene under high pressure conditions using water as the pressure transmission medium (PTM). The results are explained in terms of a breakdown in the Kohn anomaly associated with the finite size of remaining $sp^2$ sites inside the rehybridized 2D matrix. Ab initio calculations and molecular dynamics (MD) simulations are employed to clarify the formation mechanism in the present experimental conditions tested. Additional experiments performed in single-layer graphene using water as PTM, and also in double-layer graphene using mineral oil as PTM indicate that the pressure-induced formation of diamondene is drastically favored by the stacking of two or more layers of graphene surrounded by specific chemical groups such as hydroxyl groups and hydrogens.

## Results

**Raman analysis.** Figure 1a shows the schematic of the experimental setup. The sample was placed into a diamond anvil cell (DAC) capable of operating up to ≈15 GPa. The details about the experimental conditions are provided in the Methods section. Figure 1b shows the evolution of the first-order Raman-allowed

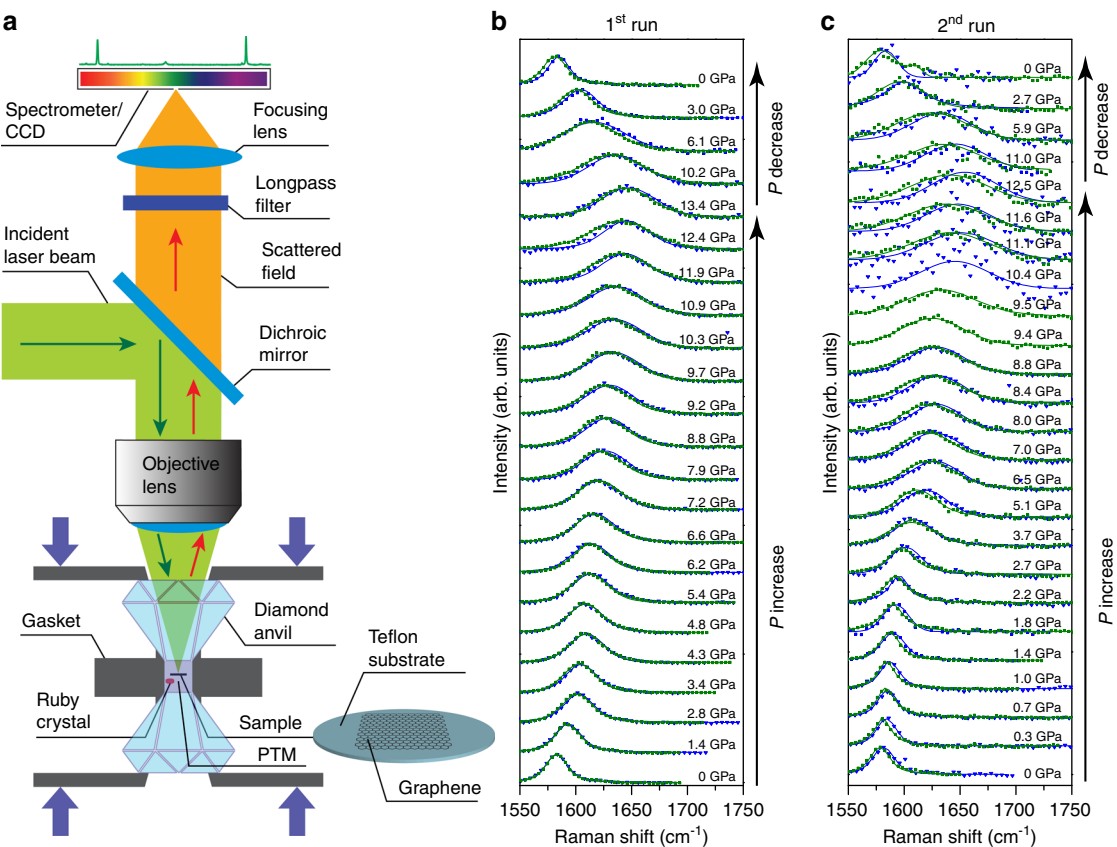

**Fig. 1** Experimental setup and Raman spectra. **a** Schematic of the experimental setup. The sample was placed into a diamond anvil cell (DAC) capable of operating up to ≈15 GPa. The details about the experimental conditions are provided in the Methods section. **b**, **c** Evolution of the G band with increasing pressure (up to ≈14 GPa) using water as PTM. The respective applied pressure is indicated on the right side of each respective spectrum. The sample used in this experiment was a double-layer graphene transferred to a Teflon substrate (G/G/T). Two spectra are shown for each pressure level, one obtained with an excitation laser energy $E_L = 2.33$ eV (*green symbols*), and the other with $E_L = 2.54$ eV (*blue symbols*). The *solid lines* are Voigt fit to the experimental data. All intensities were normalized to show approximately the same peak height. The data shown in (**b**) and (**c**) were obtained in two distinct measurement runs (first and second, respectively), performed in two distinct G/G/T samples

bond-stretching G band with increasing pressure (up to ≃14 GPa) using water as PTM. Due to the superposition of the D (~1350 cm$^{-1}$) and 2D (~2700 cm$^{-1}$) bands with the first- and second-order bond-stretching peaks of diamond, respectively, the G band was the only clearly observable Raman feature from graphene in the high-pressure experiments. The sample used in this experiment was a double-layer chemical vapor deposition (CVD)-grown graphene transferred to a Teflon substrate (G/G/T)[23]. It should be noted that what we call double-layer graphene is, in fact, a structure formed by the deposition of a single layer of graphene on top of another single layer of graphene. This is different from the traditional bilayer graphene with AB stacking.

Two spectra are shown in Fig. 1b for each pressure level, one obtained with an excitation laser energy $E_L = 2.33$ eV (green symbols), and the other with $E_L = 2.54$ eV (blue symbols). The solid lines are Voigt fit to the experimental data. A quick visual inspection of Fig. 1b reveals that the G band becomes steeper and broader as the pressure increases. These two events can be seen in detail in Fig. 2a, b, which show the plots of the G band frequency ($\omega_G$) and line width ($\Gamma_G$), respectively, as a function of the pressure ($P$), both of which were extracted from the spectra in Fig. 1b. As shown in panel 2(a), $\omega_G$ undergoes a (rough) linear blueshift with increasing pressure (filled symbols), and the change is reversible upon pressure release (empty symbols). The main cause for this dispersive behavior is a pressure-induced hydrostatic strain that generates G-phonon hardening[24, 25]. Another possible cause is the occurrence of charge transfer between the PTM and the G/G sample (the so-called pressure-induced doping), although significant doping from PTM is questionable[26, 27].

The G band broadening with increasing pressure observed in Fig. 2b (full symbols) is also reversible upon pressure releasing (empty symbols). Previous high-pressure Raman experiments

conducted on graphite indicate that this material undergoes a phase transition for pressure values between 10–20 GPa, turning into a diamond-like material in which $sp^2$ and $sp^3$ hybridizations coexist[28-30]. This $sp^2$/$sp^3$ mixed phase has been confirmed through other experimental techniques, such as inelastic X-ray scattering[31], optical transmittance[32], and electrical resistivity[33]. Several theoretical models have been proposed to explain these experimental findings[28, 31, 34-37], and the general consensus is that this diamond-like phase originates from the formation of $sp^3$ bonds, favored by the enhanced interlayer interaction induced by high pressure. High-pressure Raman experiments were also conducted in mono- and few-layer graphene samples[26, 27, 38-40] and a phase transformation has been observed in graphene nanoplates at 15 GPa[38]. This phase transformation is associated with an abrupt broadening of the G band, explained in terms of interlayer coupling that gives rise to $sp^3$ bonds in these few-layer graphene nanoplates[38]. In this work, the main contribution to the G band broadening upon compression is probably the extra strain and stress gradients caused by substrate deformation and quasi-hydrostaticity of the medium. The hydrostaticity of the water medium was inferred in our experiments by analyzing the ruby's fluorescence peaks, as shown in Supplementary Fig. 1 of Supplementary Note 1.

The presence of $sp^3$ sites in graphitic systems results in G band frequency dispersion with excitation laser energy (the frequency gets higher with increasing excitation laser energy)[41]. Accordingly, the data shown in Fig. 2a confirm that the G band frequency obtained with $E_L = 2.33$ eV and $E_L = 2.54$ eV (defined as $\omega_G^{green}$ and $\omega_G^{blue}$, respectively) splits for $P \geq 7.5$ GPa, with $\omega_G^{blue}$ getting systematically higher than $\omega_G^{green}$. The splitting can be better visualized in Fig. 2c, which shows the plot of the difference $\Delta\omega_G = \omega_G^{blue} - \omega_G^{green}$ as a function of $P$. The dashed-red line is a step-function fit to the experimental data, and the gray areas

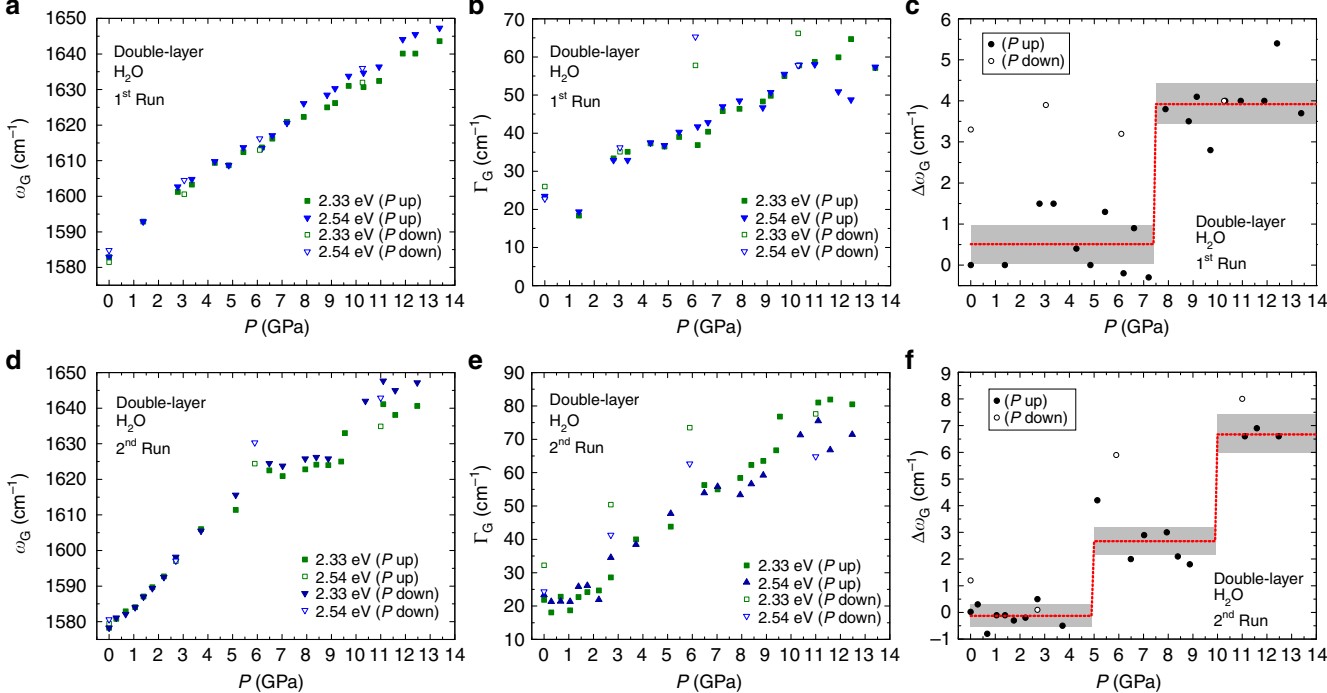

**Fig. 2** Raman parameters as function of pressure. **a–c** and **d–f** show data extracted from the first (spectra shown in Fig. 1b) and second (spectra shown in Fig. 1c) runs, respectively. Filled/empty symbols correspond to data obtained during pressure increase/decrease. **a, d** G band frequency ($\omega_G$) as a function of the pressure (P). **b, e** G band width ($\Gamma_G$) as a function of P. Green and blue symbols in panels **a, d** and **b, e** are applied for data taken with $E_L = 2.33$ and 2.54 eV, respectively. **c, f** Plot of the difference $\Delta\omega_G = \omega_G^{blue} - \omega_G^{green}$ as a function of P, where $\omega_G^{blue}$ and $\omega_G^{green}$ are the G-peak central frequencies obtained with $E_L = 2.54$ and 2.33 eV, respectively. The dashed-red line is a step-function fit to the experimental data, and the gray areas delimit the 95% confidence intervals of the fitting parameters

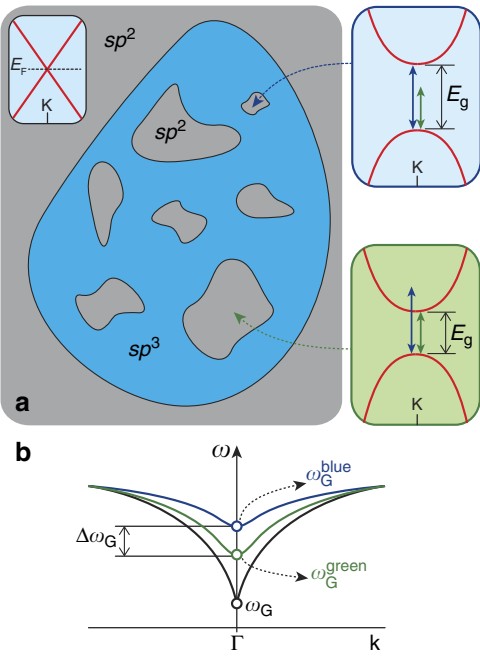

**Fig. 3** Mechanism for the G band dispersion with excitation laser energy. **a** Illustration of an $sp^3$ matrix (*blue region*) inserted in a graphitic ($sp^2$) system (*gray region*). The inset at the *left* side shows the linear energy dispersion of $\pi$ electrons at the corner (K point) of the first Brillouin zone of pristine graphene. $E_F$ stands for the Fermi level, which occurs at the K point for undoped graphene. The quantum confinement of $\pi$ electrons inside small $sp^2$ sites opens up band gaps of magnitude $E_g$ at the K point. The smaller/larger $\ell^2_{sp}$ is, the wider/narrower the associated band gap becomes (larger/smaller $E_g$), as illustrated in the top/bottom insets at the *right* side. The lengths of the *green* and *blue arrows* in these insets represent the photon energies of the blue ($E_L = 2.54\,eV$) and green ($E_L = 2.33\,eV$) excitation laser sources, respectively. An eventual match between $E_L$ and $E_g$ enhances the Raman scattered signal, and therefore smaller/larger $sp^2$ sites favor the Raman signal obtained with the blue/green laser source. **b** Double-degenerated TO/LO phonon dispersion near the center of the first Brillouin zone ($\Gamma$ point). The *black line* is related to the unperturbed graphene lattice. The $\omega_G$ is the value of the TO/LO branches at $\Gamma$, and the kink in the TO/LO dispersion is a Kohn anomaly. The presence of the band gap in the $\pi$ electron dispersion near the K point weakens the screening effect that gives rise to the Kohn anomaly, which attenuates the softening in $\omega_G$. Since the band gap becomes wider as the $sp^2$ sites become smaller, the G band measured using the blue laser presents a higher frequency value than the G band frequency measured with the green laser, that is, $\omega_G^{blue} > \omega_G^{green}$

delimit the 95% confidence intervals. A complete statistical analysis of all data presented in this work is discussed in Supplementary Note 2. Apart from the step function fitting (fitting parameters shown in Supplementary Table 1), we also performed a hypothesis test on the difference in means (detailed description presented in Supplementary Tables 2 and 3), with normality tested by the Shapiro-Wilk method (details in Supplementary Table 4) and visual inspection of Q-Q plots (shown in Supplementary Fig. 2). For pressures below 7.5 GPa, no considerable difference between the G band frequencies obtained with different laser sources is observed ($\Delta\omega_G \simeq 0.5\,cm^{-1}$), as obtained by the step-function fitting (*dashed-red line*). For $P \geq 7.5$ GPa, the step-function fitting gives $\Delta\omega_G \sim 3.9\,cm^{-1}$. It is important to notice that the splitting is irreversible upon pressure release (*empty symbols*), even for values below 7.5 GPa.

The dependence of $\omega_G$ and $\Gamma_G$ on $P$ was measured in another high-pressure Raman experiment (second run) carried out under the same experimental conditions (water as PTM), but using a distinct G/G/T sample. Figure 1c shows the G band data obtained from this second run. The fitting parameters extracted from the spectra shown in Fig. 1c are presented in Figs. 2d–f. The general trend is similar to the one observed in the first run (data shown in Fig. 1b and 2a–c), although some differences can be noted. First, the $\omega_G$ vs. $P$ plot (Fig. 2d) exhibits two plateaus, probably related to the loss of hydrostaticity of the water medium, which becomes quasi-hydrostatic in the interval 2–10 GPa (the hydrostacity of the water medium is discussed in the Supplementary Note 1 available). Second, we found $\Delta\omega_G \sim 2.7\,cm^{-1}$ in the interval 5–10 GPa, and $\Delta\omega_G \sim 6.7\,cm^{-1}$ above 10 GPa (Fig. 2f) (see details about the step-function fitting process in the Supplementary Note 2 available). Moreover, we have found that the blueshift $\Delta\omega_G$ in this second run was reversible upon pressure release, as can be seen by following the empty symbols in Fig. 2d, f. At last, the G band broadening with pressure was considerably steeper in the second run, which can be easily checked by direct comparison between the data shown in Fig. 2b, e.

The blueshift of the G band with increasing excitation energy suggests the occurrence of a system in which the $sp^2$ and $sp^3$ phases coexist. This system is idealized in Fig. 3a, which illustrates an $sp^3$ matrix (*blue region*) inserted in a graphitic system (*gray region*). This type of system involves a wide sort of different nanometer-sized $sp^2$ domains (of characteristic lateral length $\ell^2_{sp}$) with distinct electronic and vibrational properties due to quantum confinement. In this scenario, the confinement of $E_{2g}$ phonons within $sp^2$ domains that are smaller than the phonon coherence length contributes to the G band broadening (the phonon coherence length in graphene is in the order of tens of nanometers)[42, 43].

As illustrated in the inset at the left side of Fig. 3a, the gapless energy dispersion of $\pi$ electrons in pristine graphene is linear and symmetric around the corner of the first Brillouin zone (K point). The coupling of $\pi$ electrons or holes with zone-center ($\Gamma$ point) transversal and longitudinal optical phonons (TO and LO, respectively) gives rise to a strong screening effect that generates a kink (frequency softening) in the degenerated TO and LO phonon branches at $\Gamma$ point. This sudden softening is called Kohn anomaly[44], and is illustrated in Fig. 3b (*black line*). Since the G band originates from the double-degenerated/zone-center $E_{2g}$ phonon mode (TO/LO at $\Gamma$ point), its frequency is extremely sensitive to eventual changes in the oscillation strength of electron-phonon interactions near the Fermi level[44].

The quantum confinement of $\pi$ electrons inside small $sp^2$ sites opens up a band gap of magnitude $E_g$ at the K point. The smaller/larger $\ell^2_{sp}$ gets, the wider/narrower the associated band gap becomes (larger/smaller $E_g$), as illustrated in the top/bottom insets at the *right* side of Fig. 3a. The presence of this band gap weakens the Kohn anomaly effect, which attenuates the softening in $\omega_G$. At this point, we arrive at the conclusion that the smaller (larger) $\ell^2_{sp}$ gets, the higher (lower) $\omega_G$ becomes.

An eventual match of the excitation laser energy $E_L$ with the band gap energy $E_g$ enhances the Raman scattered signal due to the achievement of a resonance condition in the optical absorption. As discussed in the previous paragraph, $sp^2$ sites with smaller (larger) sizes present larger (smaller) $E_g$. In this case, the Raman signal originated by smaller (larger) $sp^2$ sites are resonantly selected by higher (lower) excitation laser energies. All these facts together lead to the conclusion that smaller (larger) $sp^2$ sites, with wider (narrower) $\pi$ electron energy band gaps, are resonantly selected by higher (lower) values of excitation laser energies, generating G band scattering with higher (lower) frequencies. Therefore, the blueshift observed in the G band

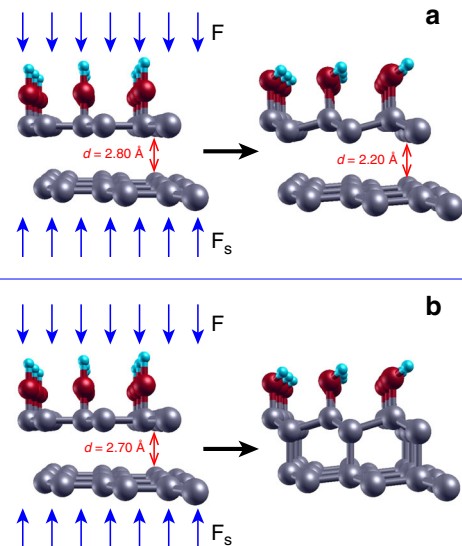

**Fig. 4** Geometries for diamondene formation. **a** Initial (*left*) and converged (*right*) geometries for the case in which the distance between graphene layers is initially set to $d = 2.8$ Å. *Blue, red,* and *gray spheres* represent H, O, and C atoms, respectively. The *vertical arrows* indicate the forces applied by the PTM (F) and the substrate underneath ($F_S$). Upon relaxation, the distance $d$ decreases to 2.2 Å, still too long to characterize a covalent interaction between layers. **b** A second calculation in which the initial distance $d$ was set to 2.7 Å. After relaxation (*right*), the diamondene forms. The C–C interlayer bond lengths become 1.66 Å, and the constrained forces are negligible

frequency for higher values of $E_L$ supports the proposition that a mixed $sp^2/sp^3$ system is formed when the double-layer graphene is subject to high pressures.

The principle underlying the diamondene formation is that the presence of chemical radicals, such as hydroxyl groups or hydrogens, may substantially decrease the pressure required to promote covalent bonds between carbon atoms in distinct layers of a double-layer graphene. For an ideal coverage of such groups, the result is a stable structure in which all carbon atoms of the upper layer are found in $sp^3$ hybridization due to the formation of C–OH or C–H bonds surrounded by three C–C interlayer covalent bonds. The presence of a substrate prevents the interaction of the lower carbon atoms with additional chemical groups. However, the carbon atoms at the bottom layer may chemically bond to the underlying substrate. In order to prevent this possibility, we used Teflon substrates in our experiment, since it is a known chemically inertial material. The final structure can be shown to be stable even in the absence of external pressure. Reversible structures may also occur if the coverage is incomplete[22].

**Theoretical analysis**. To further investigate the mechanism of diamondene formation in the present context, we have performed first principles density functional theory (DFT) calculations as well as MD simulations based on model potentials (technical details are described in the Methods section). Similar formalisms have been employed recently in studies concerning the diamondization of functionalized few-layer graphene[45–47]. In both approaches, we began with the bilayer graphene in presence of chemical groups (–H or –OH). In the DFT approach, we focused on quantitative aspects—the determination of the pressure threshold required to transform the system (either with –OH or –H groups) into the $sp^3$ network and the structural characterization of the final compound. On the other hand, the MD

simulations aimed at qualitatively describing the formation and the stability of the system (–H case) subjected to pressure at room temperature. In the model assumed in the first principles description, the pressure was imposed by geometric constraints in specific atoms during relaxation. The initial geometry was chosen with the lower C atoms and upper O atoms (–OH case) or H atoms (–H case) placed in the $z = 0$ and $z = z_0$ planes, respectively. During the relaxation, the vertical displacements of the lower C atoms were constrained to take place only in the positive $z$ direction, while the oxygen atoms of the –OH groups (or hydrogen atoms of the –H groups) were allowed to vertically move only in the negative $z$ direction. The displacements were not constrained in the $xy$ plane. When the convergence criterion was reached, the constrained forces were used to estimate the applied pressure.

Figure 4a shows initial (left) and converged (right) geometries for the case in which the distance between graphene layers was initially set to $d = 2.8$ Å. *Blue, red,* and *gray spheres* illustrate H, O, and C atoms, respectively. Upon relaxation, the distance $d$ decreases to 2.2 Å, still too long to characterize a covalent interaction between layers. Indeed, the lower layer did not present any corrugation that could indicate a deviation from the planar $sp^2$ network. The constrained forces in this final geometry correspond to an applied pressure of 4.7 GPa. On the other hand, Fig. 4b shows a second calculation in which the initial distance $d$ was set to 2.7 Å. After relaxation, the diamondene is formed, as depicted in the right side of the figure. The C–C interlayer bond lengths become 1.66 Å, and the constrained forces are negligible. The calculations were repeated for distances $d = 2.6, 2.5, 2.4$ and 2.3 Å, all of which lead to diamondene formation. Altogether, these calculations allowed us to estimate a critical pressure around 4.7 GPa.

The rehybridization process reported in the last paragraph also applies to –H chemical groups, as confirmed by MD simulations using LAMMPS package[48]. The simulations were performed for a model comprising a total of 2288 carbon atoms representing a bilayer graphene in which the upper layer interacts with 572 hydrogen atoms. External pressure was applied through two pistons modeled as purely repulsive force-field walls. The first piston was fixed and acted only on the carbon atoms of the lower layer. On the other hand, the top piston (initially localised 1.63 Å above the upper carbon atoms) acted on the whole system, being dynamically driven to reach specific levels of pressure.

Figure 5a shows a pressure vs. time ($t$) plot that summarizes the results obtained from the MD simulations. The procedure was divided into five stages, indicated in Fig. 5a and described as follows. Stage (i) corresponds to thermal equilibration, in which the system runs for 100 ps at 300 K, with pressure fluctuating around zero. The loading process takes place during stage (ii), throughout the linear approach of the top piston for 100 ps. The load achieved in the previous stage is kept constant for 2 ns (by fixing the final top piston position) in stage (iii), keeping the system in pressure equilibration. The unloading is carried out in stage (iv), when the piston is released and linearly goes back to its initial position during 100 ps. The final structure equilibration is achieved in stage (v), which takes an additional 100 ps after total piston release. Figure 5b is a zoomed version of 5(a), stressing pressure levels close to diamondene transition (between 4 and 5 GPa).

The *red curve* in Fig. 5 shows the evolution of the system when compressed up to an instantaneous (non-equilibrium) peak pressure of 4.92 GPa, indicated by the bullet in panel (b). As the system evolves in the pressure equilibration stage, the pressure slightly decays; after 0.95 ns, a transition to diamondene starts at 4.57 GPa (indicated by the circle in Fig. 5b), when a sharp pressure drop (to about 4.0 GPa) takes place. The characteristic

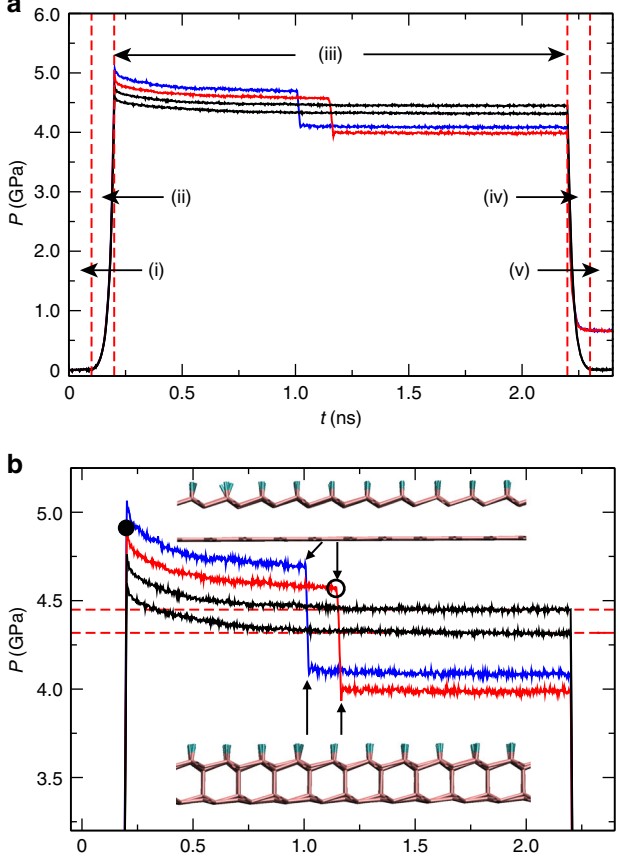

**Fig. 5** Molecular dynamics simulations. **a** Pressure versus time ($t$) curves indicating the stages imposed on the system in the simulations: (i) thermal equilibration, (ii) loading, (iii) pressure equilibration, (iv) unloading, and (v) final structure relaxation (see the text for details). *Blue* and *red* curves show cases in which diamondene transition is detected, *black* ones indicate that it is not observed. **b** Zoomed version of (**a**), stressing pressure levels close to diamondene transition (between 4 and 5 GPa). *Bullet* in the *red curve* marks the critical peak pressure necessary for $sp^3$ transition occurs (4.92 GPa). The *circle* marks the critical pressure in which the transition effectively happens (4.57 GPa). *Dashed lines* depict equilibrium pressures for graphene bilayer under piston load, i.e., when no $sp^2$-to-$sp^3$ transition is observed. The drawings illustrate the characteristic geometries just before (*top*) and after (*bottom*) the transition to diamondene takes place

expected for time periods greater than 2 ns, since the bilayer evolves without significant structural changes.

The general picture that emerges from these theoretical results is that under high pressures, as the distances from water molecules and from the adjacent layer decrease, the carbon atoms of the top layer acquire an $sp^3$ component in their hybridizations. This process increases their reactivity, making them act as dangling bond centers. Simultaneously, the highly polarized bonds in the nearby water molecules weaken upon approximation to these centers. Water molecules in contact with the top graphene layer are in crystal form (water freezes under ≈1.0 GPa at room temperature), and depending on which atom (H or O) is closer, the final result may be a mixture of C–H and C–OH bonds. Furthermore, the fact that water molecules are relatively small prevents steric-hindrance effects, allowing the formation of these bonds in multiple sites. The resulting structure, the diamondene, may be characterized as a 2D compound, which belongs to the hexagonal crystal family with lattice parameter $a = 2.55$ Å. In this regard, it is worth comparing it with the hexagonal diamond, a bulk material also known as lonsdaleite, which is focus of intense debate in the literature[49].

Lonsdaleite has a wurtzite crystal structure with interlayer bonds in the eclipsed conformation. As such, an ultra-thin compound derived from it may be viewed as the result of the compression of a bilayer graphene in the AA stacking, rather than in the AB stacking as as in the diamondene case. Our DFT calculations indicate that a lonsdaleite-diamondene is energetically less favourable by 50 meV per primitive cell when compared with the diamondene conformation described in the present work. Nevertheless, kinetic aspects may play an important role in the diamondization process as in the bulk case[50], and we cannot rule out the existence of a mixture of ultra-thin lonsldaleite and diamondene in our samples. It must be pointed out, however, that the conclusions of the present work are restricted to bilayer graphene under pressure in the presence of reactive groups, and may be extended to the two top layers of few-layer graphene[22]. The $sp^2$ to $sp^3$ transformation of the entire graphite structure is a completely different issue—it would involve either the analysis in other pressure ranges and/or the addition of catalysts on both sides of the few layer graphene, as discussed in ref. [50]. It is, therefore, not considered or discussed in the present work. Additionally, we would like to stress that further experimental investigation (e.g., X-ray and/or electron diffraction techniques) is necessary to unequivocally determine the crystal structure of diamondene. For example, X-ray diffraction of bilayer graphene under high pressure could be performed in third generation synchrotron light sources, eventually demonstrating the diamondene structure.

**The Raman cross-check**. As discussed above and experimentally explored in ref. [22], to achieve the diamodene formation within the pressure range employed in the current work, the use of water as the PTM is absolutely necessary, since it provides the chemical groups that covalently bond to the carbon atoms in the top layer, stabilizing the $sp^3$ structure. Additionally, the diamondization of single-layer graphene in water is expected to occur at much higher pressure ($P \geq 20$ GPa) than the maximum achieved in the present work. These two limitations open the possibility to test the diamondene hypothesis by simply carrying out high-pressure Raman experiments in two different systems: a single-layer graphene transferred to a Teflon substrate (G/T) using water as the PTM, and a double-layer graphene transferred to a Teflon substrate (G/G/T) using mineral oil (Nujol) as the PTM. We performed these two experiments applying the same conditions as before (for acquiring the data shown Figs. 1 and 2), and the

geometries just before and after the transition takes place are illustrated in Fig. 5b (*top* and *bottom* cartoons, respectively). The pressure value remains constant until the end of this stage, and no substantial changes can be observed in the diamondene structure, which remains stable even during the unloading stage and after the final equilibration period. A second run, depicted by the *blue curves* of Fig. 5a, b, was performed for a slightly higher peak pressure (5.06 GPa). This second run confirmed the phenomenology observed in the first one (*red curves*), with the diamondene formation taking place in a shorter time window, as expected. The residual pressure observed in both cases (first and second runs) after pressure release (stage (v)) is an artifact introduced by the fact that the simulation box is not rescaled along the periodic directions after the structural transition takes place. Additional runs were conducted in a similar fashion for peak pressures smaller than 4.92 GPa (*black curves* in Fig. 5). In this case, diamondene formation was not observed in the overall simulation time, which corresponded to 2 ns in the pressure equilibration stage. The stabilization indicates that transitions are not to be

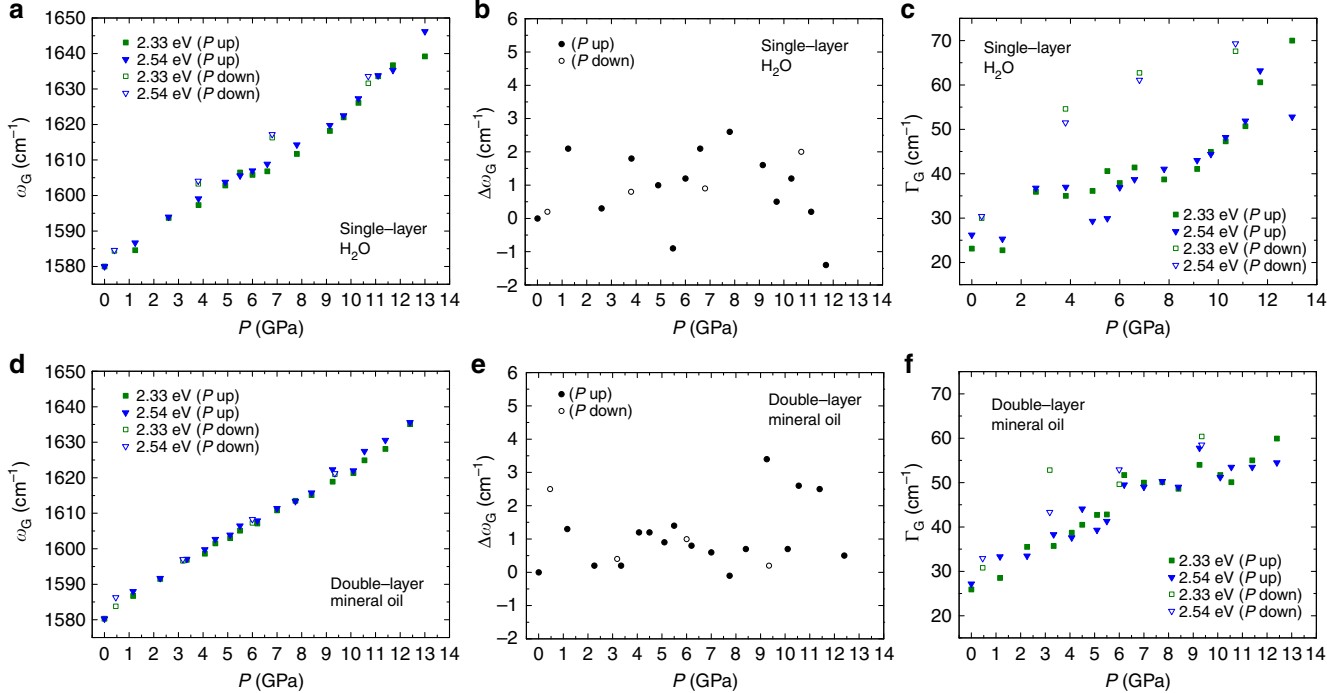

**Fig. 6** High-pressure Raman experiments performed in different systems. **a–c** Single-layer graphene transferred to a Teflon substrate (G/T) using water as PTM. **d–f** Double-layer graphene transferred to Teflon substrate (G/G/T) using mineral oil (Nujol) as PTM. The same experimental procedures as performed for acquiring the data shown in Figs. 1 and 2 were repeated in both cases, and the same notations apply

results are shown in Fig. 6. No statistically significant shift on the G band frequency with the excitation laser energy was observed (see discussion in the Supplementary Note 1 available), either for the single-layer in water, (Fig. 6a, b), or for the double-layer in mineral oil (Fig. 6d, e). These observations provide additional evidence to support the hypothesis of diamondene formation and reinforce our theoretical predictions and previous experimental results[22], thus indicating that the formation of diamondene is strongly favorable to doubly-stacked graphene compressed in the presence of chemical radicals. It is worth noticing that, even in this case, Raman spectra obtained from the double-layer graphene outside the anvil cell after pressure release (down to atmospheric pressure) indicate that the diamondene structure did not survive to ambient conditions.

## Discussion

We have provided spectroscopic evidence for the existence of diamondene by performing high-pressure Raman spectroscopy experiments in double-layer graphene using water as the PTM. The current technology of high pressure and high temperature cell apparatus involving larger volumes can make it possible to scale this novel material in a bulk quantity[51]. Potential applications include spintronics for quantum computation[52], micro-electromechanical systems (MEMS)[17], superconductivity[18], electrodes for electrochemical technologies[19], substrates for DNA-engineering[20], biosensors[5, 21], among others. Since the Raman analysis presented here provides indirect evidence for the diamondene formation, an important extension of this work would be the direct measurement of the 2D hexagonal diamond structure by X-ray or electron diffraction techniques performed under high-pressure conditions.

## Methods
**Raman spectroscopy**. Raman spectra were acquired using an alpha 300 system from WITec (Ulm, Germany) equipped with a highly linear (0.02%) piezo-driven stage, and an objective lens from Nikon (20×, NA = 0.4). Two laser lines were used:

(i) a Nd:YAG polarized laser ($\lambda = 532$ nm), and (ii) an argon laser ($\lambda = 488$ nm). The incident laser was focused with a diffraction-limited spot size ($0.61\lambda/NA$), and the Raman signal was detected by a high-sensitivity, back-illuminated CCD located behind a 600 gmm$^{-1}$ grating. The spectrometer used was an ultra-high throughput Witec UHTS 300 with up to 70% throughput, designed specifically for Raman microscopy. The measurements were performed with powers of approximately 10 and 3 mW for the 532 and 488 nm lasers, respectively. These values were chosen in order to optimize the throughput signal, which was lowered due to absorption and reflection by the DAC, without causing damage due to sample heating.

**Sample loading into the high-pressure cell**. The sample was initially cut into a strip of dimensions $\sim (0.5 \times 2)$ cm. The DAC used in this experiment was a pneumatically pressurized type. The strip with the graphene was then positioned on top of the gasket in such a way that the sample was completely covering the gasket hole. After that, the DAC was closed, resulting in the G/G/Teflon/gasket to be sandwiched between the two diamonds. The pressure was then raised up to ~4 bar, when the diamond began to deform the gasket. Because the sample was sandwiched between the diamond and the gasket, it was cut and felt inside the gasket hole. Afterwards, the pressure was released back to the atmospheric level, the DAC was opened, and the PTM and ruby were added to the gasket hole.

**Theoretical calculations**. The first-principles calculations are based on the DFT[53, 54] as implemented in the SIESTA code[55, 56]. The Kohn-Sham orbitals were expanded in a double-$\zeta$ basis set composed of numerical pseudo atomic orbitals of finite range enhanced with polarization orbitals. A common atomic confinement energy shift of 0.01 Ry was used to define the basis function cutoff radii, while the fineness of the real space grid was determined by a mesh cutoff of 450 Ry. For the exchange-correlation potential, we used the generalized gradient approximation[57], and the pseudopotentials were modeled within the norm-conserving Troullier-Martins[58] scheme in the Kleinman-Bylander factorized form[59]. All geometries were optimized until the maximum force component on any atom was less than 10 meVÅ$^{-1}$. Periodic boundary conditions were imposed, with a lattice vector in the $z$ direction large enough (22.4 Å) to prevent interactions between periodic images.

As for the MD simulations, we employed the LAMMPS package[48] with the interactions between atoms modeled through AIREBO potential[60]. All trajectories were generated in the canonical ensemble by means of the Nosé-Hoover thermostat[61, 62], responsible for keeping the average temperature in 300 K. We employed a simulation box with dimensions 55.9, 54.6 and 40 Å in the $x$, $y$ and $z$ directions, respectively, with periodic boundary conditions imposed in the $xy$ plane. Two pistons, modeled as purely repulsive force-field walls, were used to apply external pressure to the system. We used a time step of 0.25 fs.

**Data availability**. The data that support the findings of this study are available from the corresponding author upon request.

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

## Acknowledgements

The authors acknowledge Ado Jorio, Marcelo O. Aguiar, Marta P. Vidal, and Rogério Magalhães Paniago for fruitful discussions. This work was supported by CNPq, FAPE-MIG, and Rede de Instrumentação em Nano-Espectroscopia Óptica. L.G.P.M. acknowledges financial support from CNPq and the grant from Program "Fórmula Santander". A.B.O acknowledges CNPq (Grants 303820/2013-6 and 459852/2014-0). M.J.S.M and A.B.O acknowledge PROPP-UFOP (Auxlio Financeiro a Pesquisador, Grant Custeio–2016). We acknowledge computational support from LCC–Cenapad–UFMG and Cesup–UFRGS.

## Author contributions

Project planning: L.G.C; sample preparation: L.G.P.M and J.K.; High-pressure Raman measurements: L.G.P.M, A.R.P, P.T.C.F, N.F.A, A.L.A, and A.G.S.F; Ab-initio

calculations: M.J.S.M. and M.S.C.M. Molecular dynamics simulations: A.B.O. and M.J.S.M.; All authors contributed to data analysis and scientific discussions.

## Additional information

**Competing interests:** The authors declare no competing financial interests.

