## [Peer Review File · Nature Communications]

Reviewers' Comments:

Reviewer #3 (Remarks to the Author)

Dear Editor,

The paper reports optical spectroscopic measurements (Raman) of graphene mono and bi-layers as a function of pressure (in the presence or not of water). The changes in the spectra are interpreted as due to the formation of sp^3 bonding of carbon atoms, formation which is facilitated by the presence of water. The authors use these observations to claim that the initial bi-couche of graphene is transformed in a bi-couche of diamond. The possible mechanisms behind this transformation are theoretically investigated by advanced computational techniques.

The paper is clear and well written. The creation of new systems by applying pressure on graphene few-layers is an active and interesting field of research. I remark that, in spite of the claims in the abstract and in the title, the paper does not clearly demonstrate the "formation of diamondene". The paper only reports anomalous Raman data which could (possibly) be related to a change of hybridization of carbon atoms in certain regions of the sample. Moreover, the interpretation of the Raman data is reasonable, given our present understanding of these systems, however it is difficult, in the absence of further studies, to tell whether it is actually correct. Once this is said, the experiments and the resulting spectroscopic data are very interesting, come at the right time, and could possibly open new relevant outcomes. This is why I recommend this paper for publication in your journal.

Here some comments the authors might want to consider:

-Fig.1 (Raman spectra as a function of increasing pressure) is an essential piece of the present work. The authors should also show the same figure done for decreasing pressure (possibly as additional material)

-If I understand correctly, the G peak is fitted with a Voigt. Some more information could be useful to give proper interpretation of the data. Is the Voigt mainly Gaussian or Lorentzian? Is it meaningful to decompose the Voigt width into the two components Gaussian and Lorentzian? Do the two components evolve differently by increasing and decreasing the pressure?

-Although the paper is very clear and the various conclusions (as they are stated within the main text) are almost always convincing, the title/abstract are very bombastic. I suggest the authors to recalibrate the message so as not to give the impression of being overselling.

-Given that an important part of the present work is computational, the authors might possibly consider to cite previous theoretical works on a similar topic. For example:

Antipina et al., J. Phys. Chem. C 119, 2828 (2015); Munizet al. Carbon 81, 663 (2015); Kvashninet al., Nano Lett. 14, 676 (2014).

Sincerely,

Reviewer #4 (Remarks to the Author)

In their manuscript „Pressure-induced formation of diamondene“, Pimenta Martins and co-authors follow the work of Barboza et al. (Ref. 22 in the manuscript), showing the pressure-induced formation of sp³ bonds between a dual layer of graphene using water as pressure-transferring medium in a diamond anvil cell in combination with in situ Raman spectroscopy. The authors support their findings with corresponding simulation work.

While both the experimental results and the simulation work seems of high quality, I am afraid the provided insights appear only incremental in the light of previous experimental and theoretical work. Barboza et al. already demonstrate the effects of hydroxyl groups both experimentally and theoretically. Moreover, hydrogen-induced sp³-hybridization of few layers of graphene is also well known from experiment and theory (see e.g. S. Rajasekaran et al. Phys. Rev. Lett. 111, 085503 (2013)). Therefore, it is at least questionable whether the work at hand principally warrants publication in Nature Communications.

Several technical comments:

1. As this is an experimental paper, I would like to see a sketch of the experiment (sample geometry on the substrate and inside the DAC as well as propagation of the excitation lasers + line-of-sight of the spectrometer). This will provide a much better overview how the experiment is done than distributing the description as text fragments all over the text and the supplement.
2. I would expect that the formation of two-dimensional hexagonal diamond (lonsdaleite) would result in similar Raman spectra. Can this option be excluded? Could it be a mixture of cubic and hexagonal diamond structures? I only see the clear evidence for sp³ bonds but not necessarily a cubic diamond structure.
3. The authors mention another experimental run with similar sample and conditions but slightly different results. The discussion is put into the supplemental material, but should be elaborated in the main text as well. It is not clear to me why one data set should be preferred over the other.
4. It is not clear why the authors chose DFT for simulating the OH-assisted transition and simply MD for the H-assisted transition.
5. Figure 2c: How is the dashed red line defined? It does not seem to be simply the average of the data points in the different sections.
6. Figure 5: It should be clear from the caption that this figure refers to the hydrogen-initiated transition (which I figure from the main text).
7. Did the authors recover samples and see whether the “diamondene” structure survives? This will be absolutely crucial for possible applications. Please comment.

Reviewer #5 (Remarks to the Author)

Martins et al report on the transformation of double-layer graphene into 2D diamond (called “diamondene”) by means of applying high pressure under ambient temperature. While I do see the high motivation for the 2D diamond material as proposed in this study, I have a few concerns about the validity of their results and conclusions.

- The diamond structure is well known to be formed when graphite or carbon based materials are subjected to high pressures and high temperatures processing by several methods. On the other hand, inducing a hexagonal diamond structure after compressing graphite under ambient temperature is very controversial (see for example the discussion and experimental x-ray diffraction results of cold graphite under pressure in Scientific Reports 2012; 2: 520)

- The broadening of the Raman peak in figure 1 could be due to the non-hydrostatic condition induced by using water as PTM (similar PTM do have a hydrostatic limit below 10 GPa as reported in J. Phys. D: Appl. Phys. 42 (2009)) ? The broadening of the G band of graphite under pressure is also reported in Scientific Reports 2012; 2: 520, however in much lower strength. It could be that the hydrostatic condition is playing a crucial role in their experiment, which would be consistent with their own words "The main cause for this dispersive behavior is a pressure-induced hydrostatic strain that generates G-phonon hardening.^{24,25}" and "In this work, additional possible contributions to the G-band broadening upon compression are extra strain and stress gradients caused by substrate deformation and quasi-hydrostaticity of the medium"

- It is very questionable their argument that "for pressures below 8 GPa, no systematic difference between the G band frequencies obtained with different laser sources is observed ($\Delta\omega_G \approx 0$)" as shown in figure 2(c). To me it seems that there is a difference in the whole pressure range, especially at high pressures when the hydrostatic limit of water is passed. However, at this point it is not possible to conclusively say anything about this difference between the G band frequencies with different laser sources since no error-bar of the fitted data is given in figure 2, and any difference is quite small and not consistent with all of their presented pressure points. Considering this if this is one of the main experimental results supporting their conclusions, this must be addressed carefully before further evaluation.

- I am quite concerned that the only structural information about the "diamondene" is given by means of DFT and molecular dynamics simulations, since these methods do need to consider several approximations in order to optimize the electronic potential and energy of the system. While these methods could effectively support their results, I would consider that the main structural result should be obtained by experimental methods such as x-ray or electron diffraction on the graphite under pressure (see Scientific Reports 2015; 5:11812 and Scientific Reports 2012; 2: 520) and the theoretical methods could be used as support.

With all these concerns, in my opinion the main conclusions of the manuscript are not strongly supported by their experimental and theoretical results. Also, the high pressure transformation of double-layer graphene into diamond under ambient temperature is controversial (see Scientific Reports 2012; 2: 520). While their Raman results should be further analyzed and discussed in terms of the pressure transmitting medium used and the error bar of the fitted data. Further experimental data using diffraction techniques to assess the structure of the transformed material would be highly beneficial to support their conclusions.

First Referee

Referee: "The paper is clear and well written. The creation of new systems by applying pressure on graphene few-layers is an active and interesting field of research. I remark that, in spite of the claims in the abstract and in the title, the paper does not clearly demonstrate the "formation of diamondene". The paper only reports anomalous Raman data which could (possibly) be related to a change of hybridization of carbon atoms in certain regions of the sample. Moreover, the interpretation of the Raman data is reasonable, given our present understanding of these systems, however it is difficult, in the absence of further studies, to tell whether it is actually correct. Once this is said, the experiments and the resulting spectroscopic data are very interesting, come at the right time, and could possibly open new relevant outcomes. This is why I recommend this paper for publication in your journal."

We thank the Reviewer for the supportive report.

Referee: "Fig.1 (Raman spectra as a function of increasing pressure) is an essential piece of the present work. The authors should also show the same figure done for decreasing pressure (possibly as additional material)."

Answer: We thank the Referee for this suggestion. Figures 1(b,c) now include the spectra for decreasing pressure for the two experimental runs of double-layer graphene in the presence of water as PTM (from now on, these two experimental runs will be called (I) G/G/T-H₂O and (II) G/G/T-H₂O). The new Fig. 1 is displayed below. The values extracted from the fitting of the experimental data obtained in the second run are now displayed in Figs. 2(d-f).

New version of Fig. 1.

New version of Fig. 2.

Referee: “If I understand correctly, the G peak is fitted with a Voigt. Some more information could be useful to give proper interpretation of the data. Is the Voigt mainly Gaussian or Lorentzian? Is it meaningful to decompose the Voigt width into the two components Gaussian and Lorentzian? Do the two components evolve differently by increasing and decreasing the pressure?”

Answer: We thank the Referee for the questions. The reason we have used the Voigt function to fit the G peak is that the spectra obtained at high-pressures cannot be satisfactorily fitted by either a Lorentzian or a Gaussian function. We have followed the Referee’s suggestion and tried to extract systematic information about the decomposition of the Gaussian and Lorentzian components from the Voigt function used to fit the experimental data, following the analysis described in Rev. Sci. Instrum. **45**, 1369 (1974). However, none of our attempts led to a pattern that could, somehow, be linked to our experimental findings in a systematic way.

Referee: “Although the paper is very clear and the various conclusions (as they are stated within the main text) are almost always condivisible, the title/abstract are very bombastic. I suggest the authors to recalibrate the message so as not to give the impression of being overselling.”

Answer: Once more, we agree with the Referee and we have changed the title to “Pressure-induced formation of two-dimensional diamond from graphene layers: a Raman spectroscopy evidence for the diamondene”.

Referee: “Given that an important part of the present work is computational, the authors might possibly consider to cite previous theoretical works on a similar topic. For example: Antipina et al., J. Phys. Chem. C **119**, 2828 (2015); Munizet al. Carbon **81**, 663 (2015); Kvashninet et al., Nano Lett. **14**, 676 (2014).”

Answer: Following the Referee suggestion, we have included proper quotations to these references in the revised version of the manuscript.

Second Referee

Referee: "While both the experimental results and the simulation work seems of high quality, I am afraid the provided insights appear only incremental in the light of previous experimental and theoretical work. Barboza et al. already demonstrate the effects of hydroxyl groups both experimentally and theoretically. Moreover, hydrogen-induced sp^3 -hybridization of few layers of graphene is also well known from experiment and theory (see e.g. S. Rajasekaran et al. Phys. Rev. Lett. 111, 085503 (2013)). Therefore, it is at least questionable whether the work at hand principally warrants publication in Nature Communications."

Answer: Our work warrants publication in Nature Communication for the following reasons:

- (i) **The subject is of great interest.** Our work represents an important step towards the realization of diamondene, a 2D material predicted to be a ferromagnetic semiconductor with spin polarized bands.
- (ii) **The work delivers the state of the art in both experimental and theoretical studies of 2D systems.** High-pressure experiments with 2D materials are extremely challenging due to many technical reasons, most of them described along this response letter. Despite these challenges, we were able to obtain robust data whose interpretation is well supported by fundamental aspects of Raman scattering theory, and also by advanced theoretical calculations.
- (iii) **This is the first optical spectroscopic evidence for the existence of diamondene.** The Raman spectroscopy study presented in this work provides a completely new approach for the proof of the existence of this material. We also provided an explanation - in terms of the breakdown of the Kohn Anomaly - for the already well-known fact that, in the presence of sp^3 bonds, the G band disperses with laser excitation energy. This is not true for sp^2 bonds only.
- (iv) **Our results are unprecedented.** Our work has a fundamental difference from the work of S. Rajasekaran et al.: in their work; which did not involve high-pressures, they obtained evidence for the formation of a few-layer diamond material in which the carbon atoms at the top and bottom layers were chemically bonded. Therefore, this material does not exhibit the periodic array of dangling bonds at the bottom layer, which gives rise to the outstanding properties of diamondene. In our work, the fact that we used Teflon (a well-known chemically inert material) as substrate for graphene, prevents the formation of chemical bonds between the carbon atoms at the bottom layer and the substrate. This claim is supported by the fact that we obtained no evidence of diamondene formation for the G/T-H₂O sample. The fact that we need at least two layers of carbon in the presence of water is an evidence that there are no chemical bonds to the substrate, otherwise one layer would suffice. Besides that, our work adds a lot to the work of Barboza et al., in a sense that we have a better understanding of the process of diamondene formation, and the experiment points out a way of preparing a large sample, rather than only very small regions as in the experiment performed using the AFM tip for pressing graphene layers. For instance, from the DFT calculations we were able to identify the critical pressure in which diamondene can be formed using water as PTM, which greatly agrees with our $\Delta\omega_G \times P$ data. For comparison, in Barboza et al. the authors only estimated an upper bound for this critical pressure, finding 45 GPa - in the present work we brought this value to the experimental range between 4 and 14 GPa. With

these results, we now know that diamondene synthesis can be achieved in realistic conditions (this is a feasible level of pressure), which is obviously a great step towards the accomplishment of practical applications because the method is scalable. Finally, we provide an intuitive picture of the process of diamondene formation based on the reactivity of the carbon atoms at the top layer upon compression combined with the weakening of the highly polarized chemical bonds in water molecules. This understanding is also important for synthesizing diamondene, given that the usual PTMs are molecular species. From this picture, we can predict that diamondene will be formed in relatively lower pressures, whenever two (or more) layers of graphene are compressed in the presence of small, strong-polarized molecules.

Referee: “As this is an experimental paper, I would like to see a sketch of the experiment (sample geometry on the substrate and inside the DAC as well as propagation of the excitation lasers + line-of-sight of the spectrometer). This will provide a much better overview how the experiment is done than distributing the description as text fragments all over the text and the supplement.”

Answer: We thank the Referee for pointing out the missing information about the experimental setup. In order to comply with the Referee request, the schematics is now shown in Figure 1(a) (see the new Figure 1 above, shown in the answer to the first question made by the First Referee).

Referee: “I would expect that the formation of two-dimensional hexagonal diamond (lonsdaleite) would result in similar Raman spectra. Can this option be excluded? Could it be a mixture of cubic and hexagonal diamond structures? I only see the clear evidence for sp^3 bonds but not necessarily a cubic diamond structure.”

Answer: We thank the referee for this comment. It is an important point, which we address in the new version of the manuscript. The lonsdaleite, or hexagonal graphite, is a wurtzite structure. An ultra-thin diamond (in the two-layer limit) based on it could be generated upon compression of AA-stacked bilayer graphene. To investigate this possibility, we performed additional DFT calculations and found that a 'lonsdaleite-diamondene' is energetically less stable than our originally proposed structure (derived from the compression of AB-stacked bilayer graphene) by 50 meV per primitive cell. We understand, however, that this information alone is not enough to rule out the possibility of formation of lonsdaleite or a mixture of lonsdaleite and diamondene, as suggested by the referee, since kinetic arguments may play an important role in the formation process. Therefore, we included this discussion in the new version of the manuscript, adding a new reference to support the importance of the kinetics in the chemical restructuring (J. Am. Chem. Soc. 2017, 139, 2545–2548). In connection with this discussion, we note that the originally proposed diamondene structure has also hexagonal symmetry. By taking into account the extended array of sp^3 bonds, and the two (in-plane) lattice vectors that define it, the diamondene can be viewed as a different conformation (when compared to a 2D wurtzite lonsdaleite) that forms a 2D hexagonal diamond. It is important to notice that, regardless of the crystal structure, cubic or hexagonal, diamondene is predicted to be a 2D ferromagnetic semiconductor.

In order to better clarify this point, the following discussion has been added to the revised manuscript:

“The general picture that emerges from these theoretical results is that under high pressures, as the distances from water molecules and from the adjacent layer decrease, the carbon atoms of the top layer acquire a sp^3 component in their hybridizations. This process increases their reactivity, making them act as dangling bond centers.”

Simultaneously, the highly polarized bonds in the nearby water molecules weaken upon approximation to these centers. Water molecules in contact with the top graphene layer are in crystal form (water freezes under ≈ 1.0 GPa at room temperature), and depending on which atom (H or O) is closer, the final result may be a mixture of C – H and C – OH bonds. Furthermore, the fact that water molecules are relatively small prevents steric-hindrance effects, thus allowing the formation of these bonds in multiple sites. The resulting structure, the diamondene, may be characterized as a 2D compound which belongs to the hexagonal crystal family with lattice parameter $a=2.55$ Å. In this regard, it is worth comparing it with the hexagonal diamond, a bulk material also known as lonsdaleite, which is focus of intense debate in the literature (Scientific Report 2012; 2: 520). Lonsdaleite has a wurtzite crystal structure with interlayer bonds in the eclipsed conformation. As such, an ultra-thin compound derived from it may be viewed as the result of the compression of a bilayer graphene in the AA stacking, rather than in the AB stacking as in the diamondene case. Our DFT calculations indicate that a “lonsdaleite-diamondene” is energetically less favourable by 50 meV per primitive cell when compared with the diamondene conformation described in the present work. Nevertheless, kinetic aspects may play an important role in the diamondization process as in the bulk case (J. Am. Chem. Soc. **2017**, 139, 2545-2548.) and we cannot rule out the existence of a mixture of ultra-thin lonsdaleite and diamondene in our samples. It must be pointed out, however, that the conclusions of the present work are restricted to bilayer graphene under pressure in the presence of reactive groups, and may be extended to the two top layers of few-layer graphene.²² The sp^2 to sp^3 transformation of the entire graphite structure is a completely different issue - it would involve either the analysis in other pressure range and/or the addition of catalysts on both sides of the few layer graphene, as discussed in Ref. (J. Am. Chem. Soc. **2017**, 139, 2545-2548). It is, therefore, not considered or discussed in the present work.

Referee: “The authors mention another experimental run with similar sample and conditions but slightly different results. The discussion is put into the supplemental material, but should be elaborated in the main text as well. It is not clear to me why one data set should be preferred over the other.”

Answer: Both datasets are on equal footing and the only reason they were separated into main text and supplemental material was to make the main text more concise. However, we agree that the overall quality of the paper can be improved by displaying both datasets in the main text and, thus, the paper has been modified accordingly. We thank the referee for this suggestion. The raw data obtained in the second run is now included in Figure 1 and the fitting parameters extracted from this second run are now plotted in Figure 2. The new versions of Figures 1 and 2 are shown above in the answer to the first question made by the First Referee.

Referee: “It is not clear why the authors chose DFT for simulating the OH-assisted transition and simply MD for the H-assisted transition.”

Answer: We agree with the referee - the mixture of two methods was not properly explained in the original version of the manuscript. This issue is extensively addressed in the new version. Basically, all calculations used to explain the models and to extract quantitative results, such as pressure thresholds, are now performed within the first principles DFT formalism. Therefore, we performed additional DFT calculations for the H-assisted transition, including geometry relaxations for several distances between layers, and we included all these results in the new version to discuss the H- and OH-assisted transitions on equal footing. The MD simulations are, in the present version, employed only to qualitatively illustrate the formation of diamondene for

the H-assisted transition at a finite temperature. The reason why the OH-process is not addressed by MD simulations concerns the model potentials employed to describe the interactions in the MD formalism: we have very well tested parameterizations for the H-C bond formation, which is not true for OH-C bonds. The separation between quantitative results (DFT) and qualitative description (MD) was very useful to clarify the text and to remove a technical ambiguity in the pressure calculation by the MD simulation - we thank the referee for helping us in improving the paper, having this comment as basis.

To better clarify this point, the following discussion has been included in the text:

“To further investigate the mechanism of diamondene formation in the present context, we have performed first principles Density Functional Theory (DFT) calculations as well as molecular dynamics simulations based on model potentials (technical details are described in the Methods section). In both approaches, we began with the bilayer graphene in presence of chemical groups (-H or -OH). In the DFT approach, we focused on quantitative aspects - the determination of the pressure threshold required to transform the system (either with -OH or -H groups) into the sp^3 network and the structural characterization of the final compound. On the other hand, the molecular dynamics simulations aimed at qualitatively describing the formation and the stability of the system (-H case) subjected to pressure at room temperature. In the model assumed in the first principles description, the pressure was imposed by geometric constraints in specific atoms during relaxation. The initial geometry was chosen with the lower C atoms and upper O atoms (-OH case) or H atoms (-H case) placed in the $z=0$ and $z=z_0$ planes, respectively. During the relaxation, the vertical displacements of the lower C atoms were constrained to take place only in the positive z -direction, while the oxygen atoms of the -OH groups (or hydrogen atoms of the -H groups) were allowed to vertically move only in the negative z -direction. The displacements were not constrained in the xy plane. When the convergence criterion was reached, the constrained forces were used to estimate the applied pressure.”

Referee: *“Figure 2c: How is the dashed red line defined? It does not seem to be simply the average of the data points in the different sections.”*

Answer: In the original version, the dashed line was simply a guide to the eyes. Motivated by the comment of the Referee, we have now performed a fitting procedure using a step function, and the result is shown in the revised version of the manuscript (dashed lines in Figures 2(c) and 2(f) for the (I) and (II) G/G/T-H₂O samples, respectively. Additionally, we have performed an statistical analysis of the experimental data, consisting in a Hypothesis Test on the difference in means. This analysis is now included in the Supplemental Material. Also, please, see the answer to the third question made by the third Referee.

Referee: *“Figure 5: It should be clear from the caption that this figure refers to the hydrogen-initiated transition (which I figure from the main text).”*

Answer: We thank the Referee for point out the missing information in the caption of Figure 5. We have fixed this issue in the revised version of the manuscript.

Referee: *“Did the authors recover samples and see whether the “diamondene” structure survives? This will be absolutely crucial for possible applications. Please comment.”*

Answer: The sample (II) G/G/T-H₂O was recovered, and Raman spectra were acquired at two different spots on the sample outside the diamond anvil cell. A comparison of the G band before and after the experiment is exhibited in the figure below. All G band intensities were normalized to the same arbitrary value, for a better visualization.

As shown in the figure, the G band returns to its original position, with $\Delta\omega_G$ being approximately zero in both positions. It was also possible to observe an increase in the D' band (approx. 1620 cm^{-1}), which is an indicative that the presence of defects is increasing. The Raman data indicate that the diamondene structure did not survive to ambient conditions for this sample. It was not possible to recover the (I) G/G/T- H_2O sample, which was damaged during the cell opening procedure.

Third Referee

Referee: *“The diamond structure is well known to be formed when graphite or carbon based materials are subjected to high pressures and high temperatures processing by several methods. On the other hand, inducing a hexagonal diamond structure after compressing graphite under ambient temperature is very controversial (see for example the discussion and experimental x-ray diffraction results of cold graphite under pressure in Scientific Reports 2012; 2: 520)”*

Answer: We agree with the referee on the fact that the formation of hexagonal diamond upon compressing graphite is controversial and a topic of intense debate in the literature, as clearly exemplified by the discussion presented in Scientific Report 2012; 2: 520. Our conclusions concerning structural transformations are restricted, in the present paper, to bilayer graphene under compression in the presence of reactive groups, and may be extended to the two top layers of few-layer graphene, in agreement with Adv. Mat. 2011, 23, 3014–3017, quoted in the new version of the manuscript. The sp^2 to sp^3 transformation of the entire graphite structure is a completely different issue - it would involve either the analysis in other pressure range and/or the addition of catalysts on both sides of the few layer graphene, as discussed in J. Am. Chem. Soc. **2017**, 139, 2545-2548. Therefore, our results are by no means in contradiction with those reported in Scientific Report 2012; 2: 520. We made this discussion clear in the new version of the manuscript, and we thank the referee for drawing our attention to this important point.

Referee: *“The broadening of the Raman peak in figure 1 could be due to the non-hydrostatic condition induced by using water as PTM (similar PTM do have a hydrostatic limit below 10 GPa as reported in J. Phys. D: Appl. Phys. 42 (2009)) ? The broadening of the G band of graphite under*

pressure is also reported in *Scientific Reports* 2012; 2: 520, however in much lower strength. It could be that the hydrostatic condition is playing a crucial role in their experiment, which would be consistent with their own words “The main cause for this dispersive behavior is a pressure-induced hydrostatic strain that generates G-phonon hardening.^{24,25}” and “In this work, additional possible contributions to the G-band broadening upon compression are extra strain and stress gradients caused by substrate deformation and quasi-hydrostaticity of the medium”.

Answer: The hydrostaticity of the water medium was monitored in our experiments by analyzing the separation between the R_1 and R_2 fluorescence lines ($R_1 - R_2$) from the ruby crystal, and the full width at half maximum of the R_1 line (Γ_{R_1}). The analysis is now included in the Supplemental Material. The data (see figure below with (a,b) and (c,d) obtained for (I) and (II) G/G/T-H₂O experiments, respectively) suggest that the water medium is hydrostatic between 1-2 GPa, quasi-hydrostatic between 2 and 10 GPa, and non-hydrostatic above 10 GPa. This observation is supported by the results of high-pressure experiments with water by Piermarini et. al., *J. of Appl. Phys* 44, 5377 (1973). Therefore, the broadening of the G band in Figure 1 could, in fact, be assigned to the quasi-hydrostaticity of the water medium. We make it clear in the revised version of the manuscript, with the sentence “In this work, the main contribution to the G-band broadening upon compression is probably the extra strain and stress gradients caused by substrate deformation and quasi-hydrostaticity of the medium”. However, besides the loss of hydrostaticity of the medium, the confinement of the E_{2g} phonons within the sp^2 domains after the diamandene formation also generates G band broadening, and cannot be ruled out in the experiments using water as PTM.

Nevertheless, our main experimental result, which is the abrupt change in $\Delta\omega_G$ for the samples (I) and (II) G/G/T-H₂O, has no connection with the quasi- or non-hydrostaticity of the PTM because the spectra obtained for both blue and green lasers were always measured at the same spot in the sample for each pressure. Therefore, although the quasi- or non-hydrostaticity of the PTM can cause some stress and non-uniformities across the graphene sample, it cannot cause any change in $\Delta\omega_G$.

Referee: “It is very questionable their argument that “for pressures below 8 GPa, no systematic difference between the G band frequencies obtained with different laser sources is observed ($\Delta\omega_G \simeq 0$)” as shown in figure 2(c). To me it seems that there is a difference in the whole pressure range, especially at high pressures when the hydrostatic limit of water is passed. However, at this point it is not possible to conclusively say anything about this difference between the G band frequencies with different laser sources since there no error-bar of the fitted data is given in figure 2, and any difference is quite small and not consistent with all of their presented pressure points. Considering this if this is one of the main experimental result supporting their conclusions, this must be addressed carefully before further evaluation.”

Answer: We completely agree with the Referee that a most careful analysis of the experimental data was missing in the previous version of the manuscript. Since this is the most critical part of our experimental results, a systematic statistical analysis is indeed necessary. We have performed this analysis, and the procedure is explained below. (This explanation is also included in the Supplementary Material). The conclusions drawn from the statistical analysis confirm that the data is robust and therefore our interpretation and conclusions are solid. Once more, we thank the Referee for this input, which brought considerable improvement to our work.

Statistical analysis of the $\Delta\omega_G$ data:

Even though there are fluctuations in $\Delta\omega_G$ over the whole pressure range, the data can be grouped into two sets with distinct mean values [in case of Fig 2(c)], with, in each group, fluctuations occurring about the mean. For Figure 2(f), there are three distinct groups. We have analyzed the $\Delta\omega_G \times P$ data extracted from the two G/G/T-H₂O samples [(I) first and (II) second runs summarized in Figures 2(c) and 2(f), respectively] by fitting them with step functions using MATLAB®. Besides that, we performed a Hypothesis Test on the difference in means for unknown variances using the statistics software R. Finally, we applied the same procedures to analyze $\Delta\omega_G \times P$ data extracted from the G/G/T-Nujol and G/T-H₂O experiments [Figures 6(b) and 6(e), respectively]. As shown in the next lines, the occurrence of distinct values of $\Delta\omega_G$ in these two data sets is not statistically supported.

Table 1 gives the information extracted from the fit of the $\Delta\omega_G \times P$ data obtained from samples (I) and (II) G/G/T-H₂O. Here a , b and c are the fitting parameters (constant $\Delta\omega_G$ values in a well defined plateau), P is the independent variable (pressure), and R^2 is, as usual, the coefficient of determination. We fixed the critical pressures separating discontinuities in $\Delta\omega_G \times P$ data: 7.5 GPa for the sample (I) G/G/T-H₂O; 5 and 10 GPa for sample (II) G/G/T-H₂O. The values of the fitting parameters, with 95% confidence bounds shown in parenthesis, are giving in Table 1. From these parameters, it is possible to infer that the $\Delta\omega_G \times P$ data for the G/G/T-H₂O samples are well explained by a Step-Function model. The same cannot be said about the data from the G/G/T-Nujol and G/T-H₂O samples, for which a critical pressure value could not be defined.

Table 1

Sample	Adjusted function	a	b	c	R^2
(I) G/G/T/H ₂ O	a if $P < 7.5$ b otherwise	0.51 (0.04, 0.98)	3.92 (3.43, 4.42)	-	0.87
(II) G/G/T/H ₂ O	a if $P < 5$ b if $5 \leq P < 10$ c otherwise	-0.13 (-0.55, 0.29)	2.67 (2.15, 3.19)	6.67 (5.97, 7.43)	0.95

To further detect a statistically significant variation between different groups of data for a given sample, a Hypothesis Test on the difference in means for unknown variances was performed using the statistics software R. For the (I) G/G/T/H₂O sample, the $\Delta\omega_G$ data were separated into

two groups: below ($\Delta\omega_{G,1}$) and above ($\Delta\omega_{G,2}$) the critical pressure $P_{c,1,2}$. For the sample (II) G/G/T/H₂O, the data were divided into three groups: $\Delta\omega_{G,1}$ and $\Delta\omega_{G,2}$ and $\Delta\omega_{G,3}$, defined by the two critical pressures $P_{c,1,2}$ and $P_{c,1,3}$. The data extracted from the G/G/T-Nujol and G/T-H₂O samples were separated into two groups, below and above a fictitious critical pressure $P^* = 6.5$ GPa. These groups are exhibited in Table 2.

Table 2

Sample	$\Delta\omega_{G,1}$	$\Delta\omega_{G,2}$	$\Delta\omega_{G,3}$	P_c (GPa)
(I) G/G/T/H ₂ O	{0, 0, 1.5, 1.5, 0.4, 0, 1.3, -0.2, 0.9, -0.3}	{3.8, 3.5, 4.1, 2.8, 4, 4, 4, 5.4, 3.7}	-	$P_{c,1,2}=7.5$
(II) G/G/T/H ₂ O	{0, 0.3, -0.8, -0.1, 0.1, -0.3, -0.2, 0.5, 0.5}	{4.2, 2, 2.9, 3, 2.1, 1.8}	{6.6, 6.9, 6.6}	$P_{c,1,2}=5$ $P_{c,1,3}=10$
G/G/T-Nujol	{0, 1.3, 0.2, 0.2, 1.2, 1.2, 0.9, 1.4, 0.8}	{0.6, -0.1, 0.7, 3.4, 0.7, 2.6, 2.5, 0.5}	-	$P^*=6.5$
G/T-H ₂ O	{0, 2.1, 0.3, 1.8, 1, 0.9, 1.2}	{2.6, 1.6, 0.5, 1.2, 0.2, -1.4, 7, 2.1}	-	$P^*=6.5$

The $\Delta\omega_G$ data within each group are considered as obtained from independent samples. This choice assumes that the source of change in $\langle\Delta\omega_G\rangle$ is solely the pressure (the symbol $\langle\rangle$ stands for the population average). The results extracted from the Hypothesis Tests for all samples, with the Null and Alternative Hypothesis stated for each sample, are exhibited in Table 3. There, d_f is the degree of freedom, t-value, t-critical and p-value have their usual meaning, and the significance is 0.05.

Table 3

Sample	Null Hypothesis	Alternative Hypothesis	d_f	t-value, t-critical (0.05 significance), p-value	Accept/Reject Null Hypothesis
(I) G/G/T/H ₂ O	$\langle\Delta\omega_{G,1}\rangle = \langle\Delta\omega_{G,2}\rangle$	$\langle\Delta\omega_{G,2}\rangle > \langle\Delta\omega_{G,1}\rangle$	17	10.54, 1.74, 3.514×10^{-9}	Reject
(II) G/G/T/H ₂ O	$\langle\Delta\omega_{G,1}\rangle = \langle\Delta\omega_{G,2}\rangle$	$\langle\Delta\omega_{G,2}\rangle > \langle\Delta\omega_{G,1}\rangle$	6	7.19, 1.9, 1.483×10^{-4}	Reject
(II) G/G/T/H ₂ O	$\langle\Delta\omega_{G,1}\rangle = \langle\Delta\omega_{G,3}\rangle$	$\langle\Delta\omega_{G,3}\rangle > \langle\Delta\omega_{G,1}\rangle$	10	28.65, 1.81, 3.124×10^{-11}	Reject
G/G/T-Nujol	$\langle\Delta\omega_{G,1}\rangle = \langle\Delta\omega_{G,2}\rangle$	$\langle\Delta\omega_{G,1}\rangle \neq \langle\Delta\omega_{G,2}\rangle$	9	1.16, 2.26, 0.2743	Accept
G/T-H ₂ O	$\langle\Delta\omega_{G,1}\rangle = \langle\Delta\omega_{G,2}\rangle$	$\langle\Delta\omega_{G,1}\rangle \neq \langle\Delta\omega_{G,2}\rangle$	13	0.93, 2.16, 0.3683	Accept

For each Hypothesis test, the equality in the population variance of the two analyzed data sets was tested with the F test and, depending on the outcome, a suitable expression to obtain the t-value was used. For the normality analysis, we employed the Shapiro-Wilk test and an additional QQ Plot. The results of the Shapiro-Wilk test are presented in Table 4, where W is the test statistics. The Null Hypothesis for the Shapiro-Wilk test corresponds to a normally distributed

population with 0.05 of significance. The Shapiro-Wilk tests showed no evidence against the assumption that the $\Delta\omega_G$ distribution is normal for all samples. This conclusion is supported by visual inspection of the QQ Plots.

Table 4

Sample	W	p-value	Accept/Reject Null Hypothesis
(I) G/G/T/H ₂ O	0.95781	0.5301	Accept
(II) G/G/T/H ₂ O	0.93207	0.2111	Accept
G/G/T-Nujol	0.94192	0.3416	Accept
G/T-H ₂ O	0.88857	0.06381	Accept

From the results of the Hypothesis Test, we reject the Null Hypothesis at 0.05 level of significance for the (I) G/G/T/H₂O and (II) G/G/T/H₂O samples, which means that the observed changes in $\Delta\omega_G$ after the critical pressures for these samples are statistically significant and cannot be attributed to chance. For the G/G/T-Nujol and G/T-H₂O samples, we accept the Null Hypothesis at 0.05 level of significance, which means that there is no statistically significant change in $\Delta\omega_G$ after the fictitious critical pressure P*. In practice, it means that observed changes in $\Delta\omega_G$ after this pressure can be attributed to chance.

QQ plots:

In summary, from the combined information provided by the Hypothesis Test and the step function fitting, we conclude that there is a statistically significant change in $\Delta\omega_G$ for pressures above 7.5 GPa for the sample (I) G/G/T/H₂O, and above 5 GPa for sample (II) G/G/T/H₂O. The same is not observed for the G/G/T-Nujol and G/T-H₂O samples, as expected. The statistical analysis show that the changes observed in the (I) G/G/T/H₂O and (II) G/G/T/H₂O $\Delta\omega_G \times P$ data cannot be attributed to chance. Therefore, it allows inferring that the plateaus observed in Figures 1(c) and 1(f) are robust. The fluctuations in $\Delta\omega_G$ that occur before these critical pressures for the (I) G/G/T/H₂O and (II) G/G/T/H₂O samples, and for the whole pressure range for the G/G/T-Nujol and G/T-H₂O samples, could be attributed to the experimental error in determination of this variable. With regards to the non-hydrostaticity of the PTM, as explained in the 2nd question, it does not affect $\Delta\omega_G$.

Referee: "I am quite concerned that the only structural information about the "diamondene" is given by means of DFT and molecular dynamics simulations, since these methods do need to consider several approximations in order to optimize the electronic potential and energy of the system. While these methods could effectively support their results, I would consider that the main structural result should be obtained by experimental methods such as x-ray or electron diffraction on the graphite under pressure (see Scientific Reports 2015; 5:11812 and Scientific Reports 2012; 2: 520) and the theoretical methods could be used as support."

Answer: We agree with the Referee that a direct structural characterization would be a great asset. However, there are technical issues that make this type of experiment, if not impossible, unrealistic with the known facilities we have access to. High-pressure experiments with 2D materials are extremely challenging in several aspects. Starting with the loading of the sample inside the high-pressure chamber, which is a hard task due to the reduced dimensions of the chamber (approx. 100 μm). The samples need to be cut in dimensions of approximately 60 μm , and micro-manipulated to be placed inside the chamber. The substrate thickness needs to be

reduced accordingly, making the exfoliation process more challenging. We have made this task slightly easier by using the Teflon substrate (because they are soft, we can load the sample inside the chamber by pressing it against the diamond). Nevertheless, the process is not 100% effective and, in the case of Raman spectroscopy, several attempts are usually needed until we get a measurable signal. Besides the sample loading, the data collection is time consuming: the Raman spectrum for each pressure point takes at least one and a half hour to be collected (we collected approximately 20 data points in a typical experiment). There are many issues with the data collection process, such as waiting for the system to reach the desired pressure, or measuring spectra from the ruby crystals with both laser excitations lines for pressure calibration purposes, and of white light for spectra calibration. In addition, the intensity of the Raman signal usually decreases with the pressure, requiring even longer accumulation times. For unloading, the fact that the substrate is soft comes with some drawbacks, such as the sample being stuck inside the chamber and/or damaged during the removal process. The use of a rigid substrate would make this process easier, but turning the loading stage considerably harder. Most of these difficulties do not exist when working with bulk materials such as graphite. That is the reason why there so many papers reporting high-pressure experiments performed in bulk, but so few performed in truly 2D materials (single or few atomic layers) up to this date.

The difficulties to perform x-ray or electron diffraction measurements in this system are enormous. We have talked to several specialists at UFMG, UFC and MIT, and all of them are skeptical about the feasibility of this type of measurement, considering the current technological possibilities. The reason is that, to measure the x-ray signal diffracted by a double-layer material, a high-energy beam would be necessary to provide the proper condition for the occurrence of a high linear absorption coefficient at the sample plane. However, because the double-layer is located inside the diamond anvil cell, this high-energy beam would be attenuated by the first few layers of the diamond crystal, making the experiment unfeasible. It is important to notice that, regardless the crystal structure, cubic or hexagonal, diamondene is predicted to be a ferromagnetic semiconductor with spin polarized bands, which own its own justifies the study of this material.

Referee: *“With all these concerns, in my opinion the main conclusions of the manuscript are not strongly supported by their experimental and theoretical results. Also, the high pressure transformation of double-layer graphene into diamond under ambient temperature is controversial (see Scientific Reports 2012; 2: 520). While their Raman results should be further analyzed and discussed in terms of the pressure transmitting medium used and the error bar of the fitted data. Further experimental data using diffraction techniques to assess the structure of the transformed material would be highly beneficial to support their conclusions.”*

Answer: We understand the Referee’s concerns, and we have worked at the edge of our possibilities to address all of them. We have responded all his/her queries, bounded by the limits of the current technology related to high-pressure apparatuses. Our theoretical analysis is deep and technically rigorous. The Raman spectroscopy experiment performed on graphene under high-pressure represents the state of the art in this field. Following the Referee comment, we have performed an extended statistical analysis on our Raman data that excludes any doubt about the main conclusions extracted from them. The diffraction measurements required by the Reviewer could not be provided but, as explained above, this failure is solely due to technological barriers (a common issue in pioneer experimental works). Therefore, we are confident that, after reading this response letter explaining the substantial improvements made on this revised version, the Referee may change his/her impression about the robustness and great progress this paper brings to the field. We are sure that the excellence contained in this work will stimulate new approaches to move forward with the science of these ultrathin materials as well as with the development of high-pressure setups for properly assessing their structures under extreme conditions.

Reviewers' Comments:

Reviewer #3:

Remarks to the Author:

Dear Editor,

after reading the author's answer to the comments

I recommend this paper for publication in your journal.

The findings are original and very interesting even though

(as I already said) the interpretation might be debatable

(as it is always the case with new results).

I agree with the authors when they say that

"Our theoretical analysis is deep and technically rigorous.

The Raman spectroscopy experiment performed on graphene under

high-pressure represents the state of the art in this field."

Sincerely,

Reviewer #4:

Remarks to the Author:

The authors have addressed all my concerns in great detail. I think the quality and clarity of the revised manuscript have strongly improved and overall, I recommend publication.

Two remaining comments:

1) I think that the results from the recovered sample (II) are interesting, even if these suggest that the proposed "diamondene" did not survive the release to ambient conditions. I recommend adding a sentence describing this observation to the manuscript.

2) All referees questioned the clarity of the evidence for diamondene as it was stated in the original manuscript. In particular, referee 3 asked for diffraction measurements to provide unambiguous structural information. While it seems obvious that obtaining clean X-ray diffraction data from a possible 2D atomic layer of diamond seems difficult in the light of a sample that is already surrounded by diamonds that are gigantic in comparison. Nevertheless, I could not completely follow the response of the authors to referee 3. Why would a "high linear absorption coefficient at the sample plane" be required to obtain a diffraction pattern? High-energy synchrotron radiation can easily penetrate diamond anvils and readily access the structure of samples inside. Such experiments are performed all over the planet. I think the main difficulty will be to clearly distinguish the diffraction signal of the 2D material from anything that could come from the diamond anvils. Such an experiment seems certainly difficult, but may not be impossible with state-of-the-art technology.

Reviewer #5:

Remarks to the Author:

Following the resubmission by Martins et al of their manuscript report on the transformation of double-layer graphene into 2D diamond under applied pressure, I recognize the great efforts made by the authors to improve their manuscript and to respond to all criticisms of the reviewers.

The current version of the manuscript is indeed robust within the limits of the methods (experimental and theoretical) used in this study. This throughout optical spectroscopy study should stimulate other investigations to prove or disprove their findings on the new phase of diamondene under pressure.

However, I disagree with the authors when they argue that x-ray experiments under high pressure on these double-layered carbon nanotubes are unfeasible. See for example Scientific Reports 6, Article number: 37232 (2016) and Proc Natl Acad Sci U S A. 2004 Sep 21; 101(38): 13699–13702. Nowadays it is quite straightforward to perform such experiments on carbon nano materials at every third generation synchrotron sources around the world. While the lack of the of proof by a direct experimental technique to determine the crystal structure under pressure does not invalidate their result and motivation, it would be very beneficial to their study.

At this stage, I am inclined to recommend publication if they make clear in the manuscript that, as a perspective for future works, structural information at the high pressure conditions must be assessed by experimental techniques (such as X-ray diffraction) in order to provide an unquestionable prove of the appearance of the 2D hexagonal diamond structure. The information provided by DFT and molecular dynamics simulations are by no means a definitive answer on this.

First Referee

Referee: *"After reading the author's answer to the comments I recommend this paper for publication in your journal. The findings are original and very interesting even though (as I already said) the interpretation might be debatable (as it is always the case with new results). I agree with the authors when they say that "Our theoretical analysis is deep and technically rigorous. The Raman spectroscopy experiment performed on graphene under high-pressure represents the state of the art in this field."*

We thank the Reviewer for the supportive report.

Second Referee

Referee: *"The authors have addressed all my concerns in great detail. I think the quality and clarity of the revised manuscript have strongly improved and overall, I recommend publication."*

We thank the Reviewer for recommending our paper for publication in Nature Communications.

Referee: *"I think that the results from the recovered sample (II) are interesting, even if these suggest that the proposed "diamondene" did not survive the release to ambient conditions. I recommend adding a sentence describing this observation to the manuscript."*

Answer: We thank the Referee for this suggestion. We have added the following sentence in line 304 of the revised version of the manuscript:

It is worth noticing that, even in this case, Raman spectra obtained from the double-layer graphene outside the anvil cell after pressure release (down to atmospheric pressure) indicated that the diamondene structure did not survive to ambient conditions.

Referee: *"All referees questioned the clarity of the evidence for diamondene as it was stated in the original manuscript. In particular, referee 3 asked for diffraction measurements to provide unambiguous structural information. While it seems obvious that obtaining clean X-ray diffraction data from a possible 2D atomic layer of diamond seems difficult in the light of a sample that is already surrounded by diamonds that are gigantic in comparison. Nevertheless, I could not completely follow the response of the authors to referee 3. Why would a "high linear absorption coefficient at the sample plane" be required to obtain a diffraction pattern? High-energy synchrotron radiation can easily penetrate diamond anvils and readily access the structure of samples inside. Such experiments are performed all over the planet. I think the main difficulty will be to clearly distinguish the diffraction signal of the 2D material from anything that could come from the diamond anvils. Such an experiment seems certainly difficult, but may not be impossible with state-of-the-art technology."*

Answer: We agree with the Reviewer that, although difficult, this type of diffraction experiment is not impossible to be performed. We also agree that the manuscript will benefit from a clear proposal for further developments. To improve this point in the manuscript, we have added the following sentences:

- (i) Line 280: *Additionally, we would like to stress that further experimental investigation (e.g., X-ray and/or electron diffraction techniques) is necessary to unequivocally determine the crystal structure of diamondene. For example, X-ray diffraction of bilayer graphene under high-pressure could be performed in 3rd generation synchrotron light sources, eventually demonstrating the diamondene structure.*
- (ii) Line 315: *Since the Raman analysis presented here provides indirect evidence for the diamondene formation, an important extension of this work would be the direct measurement of the 2D hexagonal diamond structure by X-ray or electron diffraction techniques performed under high-pressure conditions.*

Third Referee

Referee: *“Following the resubmission by Martins et al of their manuscript report on the transformation of double-layer graphene into 2D diamond under applied pressure, I recognize the great efforts made by the authors to improve their manuscript and to respond to all criticisms of the reviewers.*

The current version of the manuscript is indeed robust within the limits of the methods (experimental and theoretical) used in this study. This throughout optical spectroscopy study should stimulate other investigations to prove or disprove their findings on the new phase of diamondene under pressure.

However, I disagree with the authors when they argue that x-ray experiments under high pressure on these double-layered carbon nanotubes are unfeasible. See for example Scientific Reports 6, Article number: 37232 (2016) and Proc Natl Acad Sci U S A. 2004 Sep 21; 101(38): 13699–13702. Nowadays it is quite straightforward to perform such experiments on carbon nano materials at every third generation synchrotron sources around the world. While the lack of the of proof by a direct experimental technique to determine the crystal structure under pressure does not invalidate their result and motivation, it would be very beneficial to their study.

At this stage, I am inclined to recommend publication if they make clear in the manuscript that, as a perspective for future works, structural information at the high pressure conditions must be assessed by experimental techniques (such as X-ray diffraction) in order to provide an unquestionable proof of the appearance of the 2D hexagonal diamond structure. The information provided by DFT and molecular dynamics simulations are by no means a definitive answer on this.”

Answer: We thank the Reviewer for the supportive report. His/her request is basically the same as the one made by the Second Referee. We agree with both Reviewers that a clear proposal for further experimental developments is crucial for obtaining undoubted proof of diamondene formation. To improve this aspect, we have added the two sentences already reproduced in the response to the Second Reviewer.